# WHEN SYMBOLS SPEAK: UNDERSTANDING LOGO TRIGGERED TEXTS IN VISION-LANGUAGE MODELS

## ABSTRACT

Vision Language Models (VLMs) have achieved impressive progress in multi-modal reasoning, yet they remain vulnerable to hallucinations where outputs are not grounded in visual evidence. In this paper, we investigate a previously over-looked setting: logo hallucination, where models generate brand names or textual content despite logos containing no visible words. Using curated splits of pure symbols, hybrids, and text-bearing logos, as well as the challenging Hard-60 sub-set, we systematically measure hallucination across leading VLMs. We further probe robustness through nine structured perturbations and show that hallucina-tions persist even under strong distortions, with occlusion exposing the sharpest weakness. Embedding-level analysis with open-weight LLaVA demonstrates that hallucination is tied to a small subset of projector dimensions, and targeted abla-tion substantially reduces errors while preserving OCR accuracy. Together, these findings reveal that VLMs often rely on symbolic priors rather than genuine glyph perception, particularly for iconic circular logos, and that projector subspaces play a decisive role in this failure mode. Our work contributes both a novel diagnos-tic lens and actionable mitigation insights, highlighting projector disentanglement and OCR-guided decoding as promising directions for building more trustworthy multimodal systems.

## 1 INTRODUCTION

Recent advances in Vision-Language Models (VLMs) have enabled multimodal systems that combine the reasoning strength of Large Language Models (LLMs) with visual perception. By projecting visual features into the textual embedding space, VLMs inherit LLM capabilities and achieve strong performance across tasks such as visual question answering, captioning, and multimodal dialogue (Liu et al., 2023; Li et al., 2023a; Bai et al., 2023; Gemini Team, 2023). However, this architectural convenience also creates a structural risk: the boundary between visual and textual signals becomes blurred. When visual embeddings are adapted to resemble tokens, VLMs may conflate symbolic shapes with written words, leading to systematic hallucination.

We investigate this phenomenon in the context of logos, a domain where symbolic marks and textual glyphs frequently coexist. Surprisingly, we find that VLMs often assert the presence of text in logos that contain no characters at all. For example, when shown a simple geometric logo, state-of-the-art VLMs output well-known brand names with high confidence, even though no textual cues are present. This observation suggests that most prevalent VLMs are not performing optical character recognition (OCR) in the conventional sense, but are instead interpolating from learned priors entangled in their projector embeddings.

To systematically study this risk, we propose a three-stage framework. *Bias Analysis.* We categorize logos into ones with pure text, hybrid, pure symbol, and stylized fonts, and measure hallucination tendencies. We further stratify logos by dominant color and shape to test for superficial biases. *Per-turbation Analysis.* We apply nine structured image manipulations (blur, flips, rotations, inversion, occlusion, sharpening) to evaluate whether hallucination can be mitigated through image transfor-mations. *Projector Diagnostics.* We analyze attention maps and perform embedding ablations, revealing that hallucination is tied to a small set of projector embedding directions, not to decoding or prompting strategies.

Across four representative VLMs (OpenAI o3, Gemini, Qwen-VL, LLaVA) and others, our experiments demonstrate that logo hallucination is consistent across models, robust to perturbations, and localizable to projector subspaces. These findings highlight both the utility and risk of modality mixing: while projector embeddings empower multimodal reasoning, they also embed spurious textual priors that undermine robustness. For instance, misinterpretation of unexpected or anomalous visual elements can significantly compromise the accuracy of image captioning systems, introducing substantial noise during vision-to-text or vision-to-audio generation. Furthermore, overfitting to learned associations between specific symbols, such as luxury brand logos and their corresponding textual descriptors, may render the VLM susceptible to adversarial manipulation. The presence of counterfeit or imitation luxury branding (e.g., stylized emblems resembling Chanel or Lancôme) may induce the VLM to erroneously infer an elevated aesthetic or socioeconomic context, thereby generating responses imbued with an inappropriately "elegant" or "premium" tone as demonstrated in our qualitative case study.

To mitigate such unintended semantic drift, it is imperative to develop mechanisms that preemptively curtail the VLM's tendency to overgeneralize from symbolic cues to broad conceptual frameworks (e.g., luxury, exclusivity, sophistication) in particular cases. Such interventions should aim to decouple superficial visual markers from their associated sociocultural connotations, thereby promoting more contextually grounded and semantically accurate output generation.

Overall, this study not only deepens our understanding of VLM vulnerabilities, but also suggests practical directions for mitigating risks in security-sensitive multimodal applications.

## 2 RELATED WORK

Large-scale Vision-Language Models (VLMs) have become the backbone of multimodal AI. Flamingo (Alayrac et al., 2022) pioneered the idea of integrating a frozen LLM with vision encoders through lightweight cross-attention layers, demonstrating strong few-shot capabilities. BLIP-2 (Li et al., 2023a) further improved efficiency by introducing Q-former as an intermediate projector, enabling vision encoders to communicate with frozen LLMs with minimal fine-tuning. Instruction-tuned models such as LLaVA (Liu et al., 2023) pushed this paradigm further by aligning vision outputs to language via curated instruction data. More recently, Qwen-VL (Bai et al., 2023) and Gemini (Gemini Team, 2023) scaled this approach to tens or hundreds of billions of parameters, showcasing impressive performance on captioning, dialogue, and visual reasoning benchmarks. Despite these advances, all these systems rely on learned projectors to map visual features into text-like embeddings, which is an architectural shortcut that may blur the boundary between modalities.

Hallucination, defined as outputs not grounded in evidence, is a well-known challenge for text-only large language models (Ji et al., 2023). In multimodal contexts, early work highlighted "object hallucination" in image captioning systems, where models over-predicted salient but absent objects (Rohrbach et al., 2018). The POPE benchmark (Li et al., 2023b) formalized this issue in VLMs, demonstrating that state-of-the-art models frequently insert spurious objects in captions. Recent surveys such as Bai et al. (2024) synthesized findings across multimodal benchmarks, noting that hallucination is especially severe when text and vision embeddings are tightly entangled.

Our work extends this literature by focusing on *triggered text in logos*, a previously overlooked setting where symbolic cues are misinterpreted as written words. This complements existing work on object hallucination by revealing a new axis of modality confusion.

## 3 METHODOLOGY

VLMs inherit the linguistic reasoning capabilities of LLMs through a trainable visual projector that maps image features into the LLM's token embedding space. To rigorously investigate this phenomenon, we first characterize its scope by examining whether and how VLMs exhibit systematic biases across different logo categories: text-free logos (e.g., Nike swoosh only), text-dominant logos (e.g., McDonald's), and hybrid logos (e.g., Chanel text with interlocking Cs). We further dissect potential visual drivers of these biases by analyzing whether specific color palettes or geometric shapes correlate with model behavior.

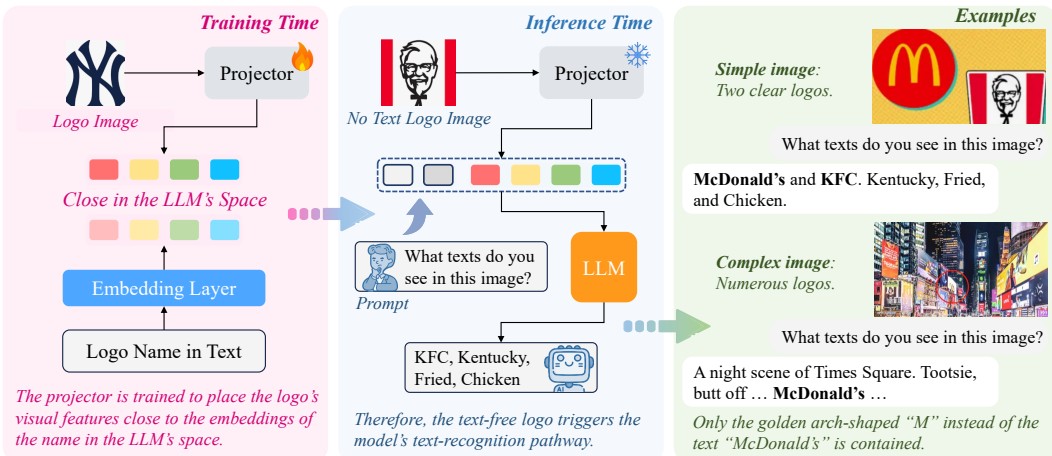

Figure 1: Logo-name leakage in VLMs. During training, the projector aligns logo visuals with their textual name embeddings. At inference, even text-free logos elicit brand-name outputs. Examples show leakage in both simple two-logo images and complex scenes with many logos and texts.

Beyond structural analysis, we conduct a qualitative case study to explore how emotionally or culturally charged logos influence semantic interpretation. For instance, models associate McDonald's with affordability and casual dining, LOUIS VUITTON with luxury and exclusivity, Nike with athleticism and motivational slogans ("Just Do It"), CHANEL with sophistication, SpongeBob SquarePants with youthfulness, and medieval coats of arms with historical gravitas. These associations reveal that VLMs do not merely read logos but interpret them through learned cultural and emotional lenses.

To uncover the root cause of such symbolic-to-textual mapping, we perform an embedding-level diagnosis, probing how the visual projector transforms logo representations into text-like embeddings that trigger specific lexical or conceptual outputs. This multi-stage investigation forms the foundation of our methodology.

### 3.1 Logo Taxonomy and Bias Characterization

First, we evaluate VLM behavior across logo categories (pure symbol, hybrid, and pure text) to quantify inherent modality-mixing bias. Following the dataset annotations and our curation, we define three canonical subsets. *Pure Symbol* indicating logos that contain only graphical or iconic elements, with no visible textual content. These are typically brand marks such as the Apple symbol or automotive badges. *Hybrid (Symbol + Text)* including logos that combine both symbolic shapes and explicit textual elements. For example, circular emblems containing both a brand wordmark and a graphic symbol. *Pure Text* consists of logos composed entirely of visible textual tokens, often stylized wordmarks such as "Google" or "Coca-Cola".

Furthermore, we curate Hard-60, a collection of 60 challenging logos, primarily consisting of cases with highly stylized fonts, cursive scripts, or decorative typefaces. Although these logos contain textual elements, the text is visually ambiguous and difficult to parse even for human annotators. This subset is designed to stress-test whether VLMs genuinely perceive glyph-level text or instead rely on learned brand priors.

To investigate whether hallucination correlates with low-level visual features such as hue, we construct a collection of logos stratified by their dominant color. Following common color groupings in branding and automotive logos, we consider six categories: black/white, silver, red, yellow, blue, and green. Each color group is sampled with an equal number of logos to avoid imbalance.

We also analyze geometric shapes as a potential confounding factor. Logos are manually grouped into four categories according to their global silhouette. Circle ∘, e.g., circular automotive emblems. Square □, logos with dominant square framing. Triangle △, logos with triangular layouts or shields. Irregular Ω, all other logos with asymmetric or complex contours.

By isolating color and shape factors, we test whether VLMs exhibit systematic bias towards specific visual appearances when hallucinating text from purely symbolic logos. Empirically, if we observe that hallucination rates remain consistently high across all categories, it suggests that the tendency to "see" text in symbols is not driven by low-level visual biases, but rather by token-level priors.

## 3.2 Perturbation Based Robustness Evaluation

To further probe the stability of hallucination under controlled input changes, we design nine perturbation types applied to logos. These perturbations cover both photometric and geometric transformations, and are intended to test whether hallucination is sensitive to superficial visual distortions. The definition of each perturbation can be found in Table 1.

Table 1: Perturbation types applied to logos. Each transformation is applied at consistent severity across the dataset.

| Perturbation | Description |
| --- | --- |
| **Blur** | Apply Gaussian blur to reduce edge sharpness and visual detail. |
| **Flip-H** | Horizontal mirroring of the logo. |
| **Flip-V** | Vertical mirroring of the logo. |
| **Invert-Color** | Replace each pixel by its color complement, reversing foreground-background contrast. |
| **Occlusion** | Mask out a random rectangular region of the logo. |
| **Rotate-180** | Rotate the logo upside down. |
| **Rotate-90** | Rotate the logo by 90° clockwise. |
| **Rotate-Randomly** | Rotate the logo by a random angle in $(0°, 360°)$. |
| **Sharpen** | Enhance edge contrast to exaggerate local features. |

Each perturbation is applied at consistent severity across the dataset, ensuring comparability across models and categories. Specifically, Gaussian blur used a kernel size up to 7×7 pixels ($\sigma \approx 3$); occlusion employed coarse dropout of up to three rectangular masks covering at most 30% of logo area each; sharpening was applied with $\alpha \in [0.2, 0.5]$ and lightness in $[0.5, 1.0]$; rotations were fixed at 90° or 180°, while random rotation sampled uniformly within $(0°, 360°)$; color inversion replaced each pixel with its complement; and flips were applied deterministically across the full image crop. The full parameterization of each perturbation is reported in Appendix Table 4, providing a reproducible specification of the exact Albumentations settings used.

## 3.3 Projector Diagnosis

To interpret the process of the triggering, we diagnose the VLM on tokens and embeddings level. By focusing attention and embedding analyses on the projector outputs, we test the hypothesis that hallucination stems from the inability of projectors to disentangle visual structure from text-like tokens. This perspective motivates future work on training projectors that better respect the boundary between symbolic and textual modalities. Since proprietary models such as Gemini-2.5 Pro and OpenAI o3 do not expose internal representations, white-box ablation is only feasible on open-weight VLMs. We therefore focus on **LLaVA-1.6**, which also triggers logo-hallucinations, and its projector activations can be directly accessed. Closed models are retained in other evaluations but are excluded here. The process is described as follows.

Let an input image produce $N$ visual tokens through the vision encoder; the projector maps them into the LLM space $Z \in \mathbb{R}^{N \times d}$ as $Z = \text{Proj}(\text{Vision}(\text{image}))$, where $d$ is the LLM embedding dimension. We form a mean pooled projector representation $\bar{z} = \text{pool}(Z) \in \mathbb{R}^d$, with $\text{pool}(Z) = \frac{1}{N}\sum_{i=1}^{N} Z_i$.

For a pure-symbol logo, let $y \in \{0, 1\}$ indicate whether the model hallucinates text ($y=1$ if hallucination occurs). Let $p_\theta(y=1 \mid \bar{z}) = \sigma(w^\top \bar{z} + b)$ with $\sigma(t) = \frac{1}{1+e^{-t}}$. We can fit an $\ell_1$-regularized logistic regression to predict $y$ from $\bar{z}$:

$$\hat{\theta} = \arg\min_{w,b} \frac{-1}{M} \sum_{m=1}^{M} \left[ y_m \log p_\theta(y_m=1 \mid \bar{z}_m) + (1-y_m) \log\left(1 - p_\theta(y_m=1 \mid \bar{z}_m)\right) \right] + \lambda \|w\|_1, \quad (1)$$

with $\lambda$ chosen to match $C=0.01$ in the implementation. The probe yields a weight vector $\hat{w} \in \mathbb{R}^d$; dimensions with large $|\hat{w}_j|$ are most predictive of hallucination. Then we rank coordinates by absolute

coefficient magnitude and keep the $k$ most predictive: $\mathcal{K}_k = \arg\mathrm{topk}_{j \in [d]} \left| \hat{w}_j \right|$ with $|\mathcal{K}_k| = k$. The value of $k$ is selected by cross-validation on a held-out set (we observe performance saturation at $k=32$). At inference, we zero the selected coordinates in every projector token before they enter the LLM, as $Z^{(\mathrm{tgt})}_{i,j} = \begin{cases} 0, & j \in \mathcal{K}_k, \\ Z_{i,j}, & \text{otherwise,} \end{cases} \quad i = 1, \ldots, N, \; j = 1, \ldots, d.$

Equivalently, with a diagonal mask $M \in \{0,1\}^{d \times d}$, $M_{jj}=0$ (if $j \in \mathcal{K}_k$ and 1 otherwise), we apply $Z^{(\mathrm{tgt})} = Z M$. As a placebo, we sample a random index set $\widetilde{\mathcal{K}}_k \subset [d]$ (uniform without replacement) and construct $Z^{(\mathrm{rnd})}$ analogously. To evaluate the change, we define the deltas of metrics. Let $\mathrm{Acc}_{\mathrm{text}}$ denote exact-match accuracy on pure-text logos and $\mathrm{Hall}_{\mathrm{sym}}$ denote hallucination rate on pure-symbol logos. For condition $c \in \{\mathrm{base}, \mathrm{tgt}, \mathrm{rnd}\}$, we calculate $\Delta\mathrm{Acc}^{(c)}_{\mathrm{text}} = \mathrm{Acc}^{(c)}_{\mathrm{text}} - \mathrm{Acc}^{(\mathrm{base})}_{\mathrm{text}}$ and $\Delta\mathrm{Hall}^{(c)}_{\mathrm{sym}} = \mathrm{Hall}^{(c)}_{\mathrm{sym}} - \mathrm{Hall}^{(\mathrm{base})}_{\mathrm{sym}}$. We report $\Delta$ relative to baseline, with targeted ablation expected to yield $\Delta\mathrm{Hall}^{(\mathrm{tgt})}_{\mathrm{sym}} < 0$ and only small $\Delta\mathrm{Acc}^{(\mathrm{tgt})}_{\mathrm{text}}$.

When probe training is infeasible, we select $\mathcal{K}_k$ by the largest mean absolute activations across tokens: $\mathcal{K}_k = \arg\mathrm{topk}_j \left( \frac{1}{N} \sum_{i=1}^{N} |Z_{i,j}| \right)$.

Overall, this method not only diagnoses the vulnerability of current VLMs but also provides a principled way to measure how visual-textual entanglement emerges in practice. By explicitly testing across logo categories, perturbation types, and projector embeddings, we lay the foundation for projector designs that are robust to hallucination and safer against malicious content insertion.

Table 2: Performance in detecting genuine text across logo overall categories, colors, and shapes for the four selected models. Distribution values are computed per model based on test set composition.

| Model→ | | OpenAI o3 | Gemini-2.5 | LLaVA-1.6 | Qwen3-VL |
|---|---|---|---|---|---|
| Category↓ | Type | Accuracy (%) | | | |
| Pure Symbol | | **47.29** | 45.69 | 33.87 | 44.39 |
| Hybrid | Overall | 96.59 | **96.98** | 80.05 | 95.41 |
| Pure Text | | **99.80** | **99.80** | 99.50 | **99.80** |
| Hard-60 | | 96.67 | **98.33** | 93.33 | 86.67 |
| | | Accuracy + Distribution (%) | | | |
| Black/White | Color ■ | **48.00** 16.285% | 47.45 16.596% | 35.84 16.623% | 46.99 16.642% |
| Silver | Color ▪ | **49.04** 16.638% | 46.78 16.362% | 36.03 16.712% | 47.33 16.762% |
| Red | Color ▪ | 48.10 16.319% | **49.06** 17.159% | 36.03 16.712% | 47.45 16.805% |
| Yellow | Color ▪ | **50.13** 17.008% | 47.78 16.712% | 35.84 16.623% | 46.99 16.642% |
| Blue | Color ▪ | **50.00** 16.964% | 47.45 16.596% | 35.84 16.623% | 46.99 16.642% |
| Green | Color ▪ | 49.48 16.787% | 47.39 16.575% | 36.02 16.707% | 46.61 16.507% |
| Circle | Shape ○ | 44.93 23.700% | 44.59 23.944% | 34.20 24.086% | 44.82 23.962% |
| Square | Shape □ | **48.81** 25.746% | 46.68 25.066% | 35.91 25.291% | 47.24 25.255% |
| Triangle | Shape △ | **46.85** 24.985% | 46.54 24.950% | 35.80 24.985% | 46.68 24.936% |
| Irregular | Shape Ω | **48.99** 24.713% | 48.42 24.991% | 36.08 25.213% | 48.31 24.956% |

## 4 EXPERIMENTAL VERIFICATIONS

We verify our analysis by evaluating two tiers of vision-language models. Our primary experiments focus on four representative models: OpenAI o3, Gemini-2.5 Pro (Gemini Team, 2023), LLaVA-1.6 13B (Liu et al., 2023), and Qwen3-VL-235B-A22B (Bai et al., 2023). These span both proprietary and open-weight families, balancing state-of-the-art closed deployments with transparent research models. All ablation and projector-space analyses are restricted to the open-weight model LLaVA-1.6 13B. We use the **LogoDet-3K** dataset (Wang et al., 2020), which provides diverse and realistic logo samples. Logos are grouped into three canonical categories (pure text, hybrid, and pure symbol). For particular verifications, we stratify specific logos from the dataset to construct Hard-60 of stylized fonts, groups of different colors, groups of different shapes, and apply the nine perturbations.

We evaluate models under two complementary criteria. For pure text logos, we calculate the *exact text accuracy* to measure OCR fidelity for logos containing genuine text. Given $N_{\mathrm{text}}$ samples with visible string labels $t_i$ and model predictions $\hat{t}_i$, we compute $\mathrm{Acc}_{\mathrm{text}} = \frac{1}{N_{\mathrm{text}}} \sum_{i=1}^{N_{\mathrm{text}}} \mathbb{1}\left[\hat{t}_i = t_i\right]$. For pure symbol

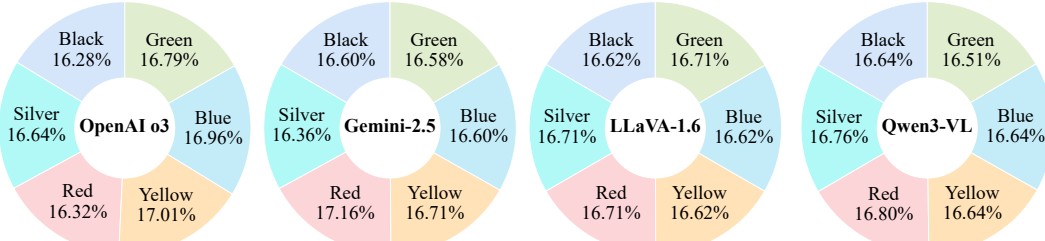

Figure 2: Donut charts illustrating the relative distribution of six color categories for each model. The percentages are computed by normalizing within each model, so they represent the ratio of one color compared to the six-color total, rather than absolute contributions to the dataset. This representation makes clear that the models exhibit no systematic bias toward any specific color.

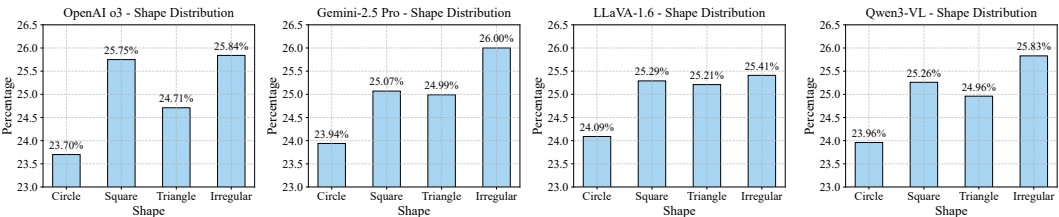

Figure 3: Shape-wise accuracy rates across four models. Although the distributions are numerically close (23-26%), a consistent trend emerges: models hallucinate most on circular logos and least on irregular-shaped logos.

or hybrid logos, we calculate *no hallucination rate* to measure the frequency with which a model does not output spurious text when none is visible. Formally, for a set of $N_{sym}$ pure-symbol logos with ground-truth label $y_i = 0$ (no text present) and binary prediction $\hat{y}_i \in \{0, 1\}$ indicating whether the model emitted textual tokens, we define hallucination rate as Hall $= \frac{1}{N_{sym}} \sum_{i=1}^{N_{sym}} 1[\hat{y}_i = 1 \wedge y_i = 0]$, so that no hallucination rate is $1 -$ Hall.

This dual metric separates two distinct capabilities: (i) avoiding false positives on symbol-only logos, and (ii) producing exact matches for glyph-bearing logos. High-quality VLMs must jointly minimize hallucination while preserving recognition accuracy.

All experiments are conducted on an NVIDIA A100 (80GB). Inference is run with greedy decoding unless otherwise stated. For each condition (logo type, perturbation), we sample 50-200 logos to balance statistical reliability with computational cost.

### 4.1 BIAS ACROSS TYPES

Across models, we consistently observe that when presented with pure symbol logos, VLMs strongly tend to output brand names as if they were visible text. For example, given the McDonald's golden arches without any accompanying letters, models output "McDonald's" with high certainty, often insisting that the text is visible. This reveals a fundamental inability to disentangle visual recognition of symbols from text extraction. We attempt strict prompt engineering (e.g., "extract only text you can visually read") as well as OCR-constrained decoding (restricting the output vocabulary to OCR-detected tokens). Both strategies fail: models persistently generate brand names not present in the visible text. The failure suggests that the hallucination is not merely a decoding issue but rooted in token-level modeling biases.

For logos with cursive or stylized fonts (e.g., Michael Anthony's Pizza), models frequently produce clean, canonical brand names, even when the underlying text is unreadable. This demonstrates that VLMs leverage prior token associations rather than true perception of stylized glyphs.

We stratify pure symbol logos by dominant color (black/white, silver, red, yellow, blue, green). We observe no significant color preference in hallucination frequency. Similarly, grouping logos by shape

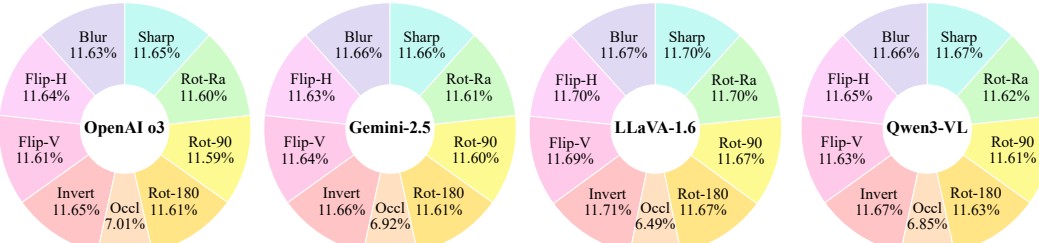

Figure 4: Donut charts illustrating the distribution of nine perturbation categories for each evaluated VLM. Percentages are normalized within each model, so values represent the relative share of errors attributable to each perturbation type. Across models, occlusion consistently accounts for the largest proportion of failures, indicating that masking critical regions is particularly detrimental for reliable logo recognition.

(circle, square, triangle, irregular) shows no systematic bias. As shown in Table 2, hallucination rates remain high across all categories and the distributions are remarkably tight (each shape localized in the 23-26% range), yet subtle and consistent trends appear across all four models as shown in Figure 3. Circle-shaped logos incur the highest hallucination rates, whereas irregular-shaped logos exhibit the lowest. A plausible explanation is that circular designs are disproportionately common among highly recognizable brands (e.g., automotive companies like Mercedes), making VLMs more prone to prematurely emit brand names based on visual priors. Conversely, irregular logos lack such entrenched associations, leading models to respond more conservatively. This analysis suggests that hallucination is not purely stochastic but can be modulated by entrenched priors tied to global logo design conventions. These results imply that hallucination is not tied to low-level visual features such as hue or geometry.

## 4.2 Perturbation Analysis

We report the two complementary outcomes, accuracy on text logos ($Acc_{text}$), and no hallucination rate on symbol/hybrid logos ($1 - Hall$). As either metric increases, the model's recognition fidelity improves and its propensity to generate hallucinated outputs diminishes. Table 3 and Figure 4 summarize the results.

Table 3: Perturbation results (accuracy on text logos; no hallucination on symbol/hybrid) on text logos across four representative VLMs.

| Model | Blur | Flip-H | Flip-V | Invert-C | Occlusion | Rot-180 | Rot-90 | Rot-Rand | Sharpen | Total |
|---|---|---|---|---|---|---|---|---|---|---|
| OpenAI o3 | 98.0% | 98.1% | 97.9% | 98.2% | 59.1% | 97.9% | 97.7% | 97.8% | 98.2% | 97.95% |
| Gemini-2.5 | 99.1% | 98.8% | 98.9% | 99.1% | 58.8% | 98.7% | 98.6% | 98.7% | 99.1% | 98.85% |
| LLaVA-1.6 | 91.0% | 91.2% | 91.1% | 91.3% | 50.6% | 91.0% | 91.0% | 91.2% | 91.2% | 91.09% |
| Qwen3-VL | 97.3% | 97.2% | 97.1% | 97.4% | 57.2% | 97.1% | 96.9% | 97.0% | 97.4% | 97.15% |

Gemini-2.5 Pro, OpenAI o3, and Qwen3-VL-235B-A22B maintain extremely high robustness ($\geq$ 97%) across nearly all perturbations. Notably, Gemini-2.5-Pro achieves 99.1% accuracy under both blur and sharpen, indicating that edge-preserving or edge-destroying distortions do not significantly disrupt its glyph recognition. This stability suggests these models rely on strong internal OCR priors rather than brittle pixel templates. LLaVA-1.6 13B exhibits clear brittleness, suggesting weak OCR alignment and underlining the gap between frontier multimodal models and more lightweight or less tuned systems. Rotations remain a universal stressor, slightly reducing text accuracy for top-tier systems, reflecting limited rotational invariance in current projector-LLM pipelines.

While most perturbations contribute nearly uniformly to error rates, occlusion emerges as a clear outlier. As shown in Table 3, occlusion consistently occupies the largest share of perturbation-induced failures across all evaluated models. This suggests that when key parts of a logo are masked, VLMs are less able to rely on their usual visual-to-text priors and are instead prone to misattributing the brand identity. Unlike color inversion or flips, which preserve global structural

cues, occlusion disrupts the most informative sub-regions of a logo, such as iconic shapes or emblematic subcomponents. This finding highlights that VLM hallucination is not purely driven by global priors but is sensitive to localized disruptions, making occlusion a particularly challenging perturbation that exposes vulnerabilities in projector-level representations.

Furthermore, the relatively uniform drop across diverse architectures implies that this vulnerability is not model-specific but inherent to the projection-based architecture common to modern VLMs. The visual projector maps entire image patches into a textual embedding space. Therefore, when critical spatial relationships are disrupted by occlusion, the resulting embeddings fail to activate the appropriate semantic clusters. This contrasts with natural object recognition, where partial views may still yield plausible interpretations via context or part-whole reasoning.

In summary, perturbations confirm that hallucination is not merely a surface-level artifact but rooted in embedding priors. The strongest models largely preserve OCR-like accuracy under diverse corruptions. Importantly, no perturbation reliably eliminates hallucinations, highlighting that mitigation requires architectural or projector-level intervention rather than data augmentation alone.

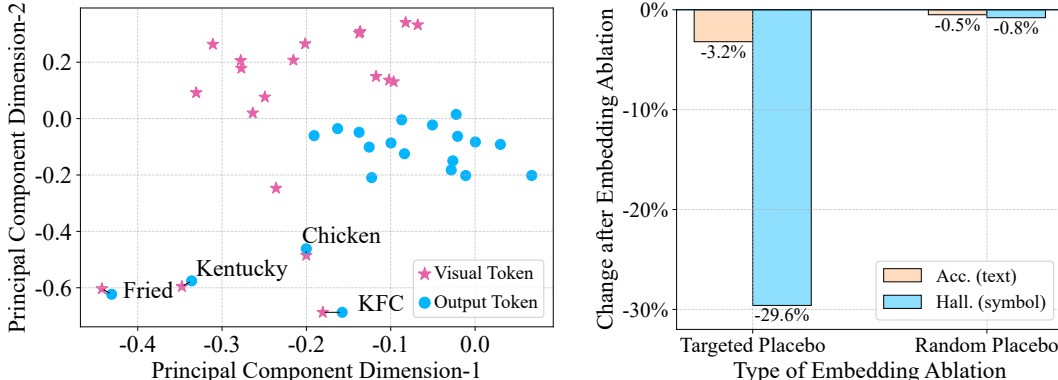

Figure 5: Embedding-level diagnosis of logo hallucination. (a) PCA projections reveal that hallucinatory outputs cluster near logo-like visual tokens, supporting the hypothesis that symbolic cues act as text triggers. (b) Ablation analysis shows that hallucination is concentrated in a small subset of projector dimensions: removing them reduces hallucination by nearly 30% while only slightly lowering text-logo accuracy.

### 4.3 EMBEDDING-LEVEL PROJECTOR DIAGNOSIS

We next investigate whether logo hallucination is localized to specific directions within the projector embedding space. Following the formalism in Section 3.3, we use the sparse logistic regression probes on pooled projector outputs to predict hallucination occurrence, and then ablate the top-$k$ most predictive coordinates. The results are shown in Figure 5.

The logistic probe assigned strongly positive coefficients to a small subset of dimensions, indicating that hallucination is not uniformly distributed across the embedding space but rather concentrated in specific sub-directions. This is further visualized in Figure 5, where we employ Principal Component Analysis (PCA) to project the token embeddings into a two-dimensional space (principal component dimension 1 and 2), enabling visual inspection of clustering patterns and semantic proximity. The projection shows that hallucinatory output tokens (e.g., "KFC") cluster near certain visual token embeddings despite no visible text in the input image. Such alignment supports the hypothesis that text-token priors are spuriously activated by logo-like features.

The targeted ablation of the top-$k$ coordinates yields a substantial reduction in hallucination ($-29.6\%$ absolute on pure-symbol logos for $k{=}32$), while accuracy on genuine text logos drops only moderately ($-3.2\%$). In contrast, the random-dimension placebo produces negligible changes ($< 1\%$), confirming that the effect arises from structured feature suppression rather than generic noise.

These findings provide concrete evidence that logo hallucination is tied to a low-dimensional subspace of the projector output. By isolating and suppressing these features, hallucination can be reduced without significantly harming OCR fidelity. This suggests that projector-level regularization, rather than naive data augmentation, may be a promising direction for mitigating multimodal hallucinations.

## 4.4 DISCUSSION

Overall, our experiments reveal three key insights: (i) VLMs cannot reliably separate visible text from brand recognition; (ii) hallucination persists even under explicit instruction and constrained decoding; and (iii) embedding-level interventions may provide a path to mitigating this issue. These findings highlight a structural limitation in current multimodal token prediction: models are overly "enthusiastic" to output canonical brand tokens whenever symbolic cues are present, undermining their trustworthiness for fine-grained text extraction.

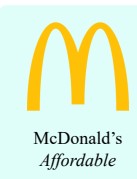 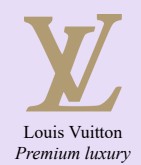 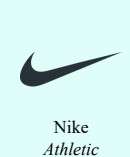 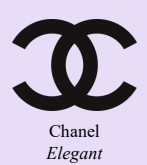 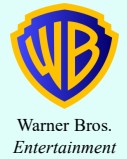 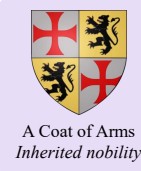

| McDonald's | Louis Vuitton | Nike | Chanel | Warner Bros. | A Coat of Arms |
| *Affordable* | *Premium luxury* | *Athletic* | *Elegant* | *Entertainment* | *Inherited nobility* |

Figure 6: Examples of logos illustrating emotional and symbolic triggering. These associations highlight how VLMs can hallucinate along value/emotional axes and how iconic shapes (e.g., the golden arch) can trigger brand recognition even in the absence of text.

## 5 LIMITATIONS AND BROADER IMPACT

Logos communicate not only textual or symbolic content but also values and emotions, such as affordability (McDonald's), luxury (Louis Vuitton), athleticism (Nike), or elegance (Chanel). Figure 6 illustrates how such associations could influence VLM behavior. Beyond text hallucination, VLMs may project branding or emotional priors onto images, potentially amplifying stereotypes or marketing signals. Similarly, iconic hand-drawn shapes (e.g., the "golden arches") can directly trigger brand recognition even without accompanying text, reflecting strong visual-text alignments from training. These observations broaden the scope of hallucination and motivate future auditing of value/emotion hallucinations, not just textual ones.

Logos provide a clean stress test where symbolic marks and textual glyphs can be disentangled, while extending projector-level interventions to natural scenes, documents, and video remains for future work. Our hallucination metric is well suited to logos, with ambiguity in stylized cases mitigated through OCR–human dual labeling. Probing and ablation identify projector directions strongly associated with hallucination, offering actionable insight even if not a full causal proof. Evaluating a representative set of VLMs, we emphasize stable relative patterns across versions, supported by standardized prompts and perturbations. Overall, the work delivers diagnostics and intervention pathways, highlighting projector disentanglement and OCR-gated architectures as promising steps toward safer multimodal systems.

## 6 CONCLUSION

Our work examined the overlooked phenomenon of logo hallucination in Vision Language Models. By analyzing pure symbols, hybrids, and text-bearing logos, we showed that models frequently output spurious text and that this behavior persists under structured perturbations such as flips, rotations, and blurring. Occlusion emerged as the strongest disruptor, while projector probing and ablation revealed that a small number of embedding directions are strongly tied to hallucination and can be suppressed with minimal impact on accuracy. These results suggest that current VLMs rely on symbolic priors rather than faithful glyph perception, especially for iconic circular logos. Although our study is grounded in logos, the findings extend to broader multimodal contexts where text and symbols co-occur. To advance reliability, future work should explore projector disentanglement and OCR guided decoding as principled solutions. By making hallucination measurable, characterizing its robustness, and diagnosing its embedding-level roots, this study offers both diagnostics and actionable interventions for building more trustworthy multimodal systems.

## ETHICS STATEMENT

We adhere to the ICLR Code of Ethics. Our work uses publicly available logo images (or indices pointing to them), and we release only indices and code (not copyrighted logos) to respect legal and ethical constraints. We analyze model failures and mitigation strategies, but we do not deploy adversarial examples intended for malicious use. We acknowledge potential misuse: insights from our work could be used to craft logos that trick VLMs, so we publish mitigation baselines and guidelines rather than releasing attack-ready artifacts. Finally, we commit full responsibility for all content, including any text aided by language models.

## REPRODUCIBILITY STATEMENT

We strive for full reproducibility. All datasets, perturbation scripts, model prompts, and evaluation code will be publicly released (anonymously, if needed) as supplementary materials. Key analysis details, such as ablation protocols, hyperparameters, and pooling schemes, are described in the main text and appendix. Any random seeds or splits used in experiments will be documented. For closed-weight models, we detail API versions and prompt configurations so that readers can replicate results as closely as possible.

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

## A  APPENDIX A: LLM USE DECLARATION

Large Language Models (ChatGPT) were used exclusively to improve the clarity and fluency of English writing. They were not involved in research ideation, experimental design, data analysis, or interpretation. The authors take full responsibility for all content.

## B  APPENDIX B: MORE ON EXPERIMENTAL SETTINGS

### B.1  PERTURBATION PARAMETERIZATION

For reproducibility, we provide the exact Albumentations configuration used to generate all nine perturbations. All transformations were implemented using the Albumentations library (version 1.4.3). The parameters below match our released code and augmentation logs, ensuring that results can be fully replicated.

Table 4: Perturbation types and parameters applied to logos. Each transformation is applied at consistent severity across the dataset (see §4.3 for analysis of results).

| Perturbation | Parameter settings |
|---|---|
| Blur | Gaussian blur, kernel size up to $7 \times 7$ ($\sigma \approx 3$). |
| Flip-H | Horizontal mirroring, $p = 1.0$. |
| Flip-V | Vertical mirroring, $p = 1.0$. |
| Invert-color | Pixel-wise complement (RGB $\mapsto$ 255 - value), $p = 1.0$. |
| Occlusion | CoarseDropout with up to 3 rectangular holes, each max 30% height/width, fill=0 (black). |
| Rotate-180 | Deterministic rotation by 180°. |
| Rotate-90 | Deterministic rotation by 90° clockwise. |
| Rotate-randomly | Random rotation sampled uniformly in $[-45°, +45°]$. |
| Sharpen | Sharpen filter with $\alpha \in [0.2, 0.5]$, lightness $\in [0.5, 1.0]$. |

## C  APPENDIX C: EMBEDDING ABLATION STUDY AND CALIBRATION ANALYSIS ON SYMBOL LOGOS

Table 5: Calibration metrics for pure-symbol logos. Lower is better for both metrics. $n$ = number of samples.

| Model | ECE | Brier | $n$ |
|---|---|---|---|
| LLaVA-1.6 baseline | 0.124 | 0.196 | 200 |
| LLaVA-1.6 targeted $k = 32$ | 0.087 | 0.158 | 200 |

## D  APPENDIX D: HARD-60 EXEMPLARS AND BUCKET CONSTRUCTION

### D.1  VISUAL EXEMPLARS WITH ANNOTATOR NOTES

**Annotation protocol (summary).**  Two trained annotators labeled each logo as *pure-symbol*, *hybrid*, or *text*, with disagreements resolved by adjudication. For *Hard-60*, we selected symbol-only logos that were repeatedly claimed as containing text across at least one VLM. For each exemplar, annotators recorded dominant color and global shape (definitions in Table 6).

### D.2  BUCKET CONSTRUCTION: COLOR AND SHAPE

**Implementation notes.**  We compute the dominant color on the cropped logo region (post-mask), using K-means ($K$=3) over HSV pixels; the majority cluster's mean hue determines the bucket. In RGB space, color categories overlap and depend on brightness/contrast. HSV separates color (Hue) from intensity (Value) and vividness (Saturation). For global shape, we take the largest contour, apply Douglas-Peucker simplification ($\varepsilon$=0.02 of perimeter), and categorize by vertex count and

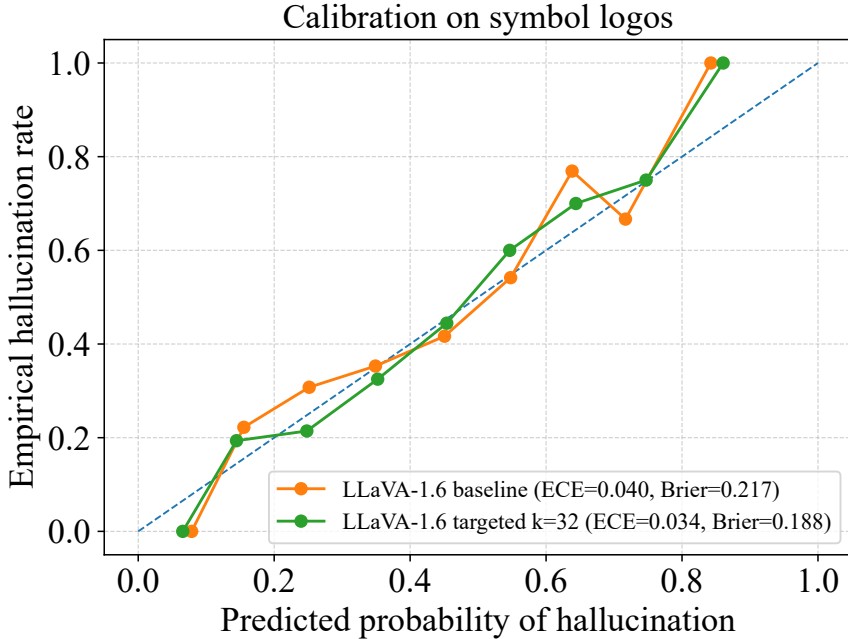

Figure 7: Calibration on pure-symbol logos. Reliability curves show empirical hallucination rate vs. predicted probability in 10 bins. Legend reports Expected Calibration Error (ECE) and Brier score per model.

Table 6: Color and shape bucket construction. Dominant color is computed from HSV hue after K-means on pixel colors within the logo crop (K=3); we take the largest cluster by area and map its mean hue to one of six bins. Global shape is assigned via contour approximation and vertex analysis (largest contour).

| Bucket | Operational definition (deterministic rule) |
|---|---|
| **Color: Green** | Hue in $[75°, 165°]$ (HSV); saturation/value $> 0.2$. |
| **Color: Black** | Value $< 0.2$ (HSV), regardless of hue. |
| **Color: Gray/Silver** | Saturation $< 0.2$ and Value $\geq 0.2$ (HSV), regardless of hue. |
| **Color: Yellow** | Hue in $[15°, 75°]$. |
| **Color: Blue** | Hue in $[165°, 255°]$. |
| **Color: Red** | Hue in $[345°, 360°] \cup [0°, 15°]$ (HSV); saturation/value $> 0.2$ to avoid grayscale; else defer to next cluster. |
| **Shape: Circle/Ellipse** | Largest contour's minimum-enclosing ellipse explains $\geq 90\%$ of contour area; aspect ratio $< 1.5$. |
| **Shape: Triangle** | Polygonal approximation (Douglas-Peucker, $\varepsilon=0.02 \times$ perimeter) yields exactly 3 vertices. |
| **Shape: Rectangle/Square** | Polygonal approximation yields 4 vertices; pairwise angles $\approx 90°$ (tolerance $\pm15°$). |
| **Shape: Complex/Polygon** | Any other convex/concave polygonal form (5+ vertices) or shape failing the above categories. |

angle heuristics. Ambiguous cases (e.g., near-square ellipse) are adjudicated by annotation, then fixed as reference labels for all experiments.

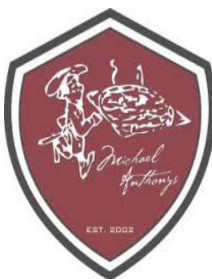

Figure 8: ID: 1
Dominant color: **Red**; Shape: **Triangle**.
GT text: *Michael Anthony's. EST. 2002*.
Annotator note: stylized curve; legible glyphs.

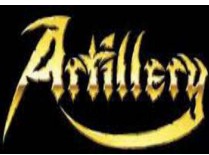

Figure 9: ID: 2
Dominant color: **Black**; Shape: **Rectangle/Square**.
GT text: *agnès b.*
Annotator note: cursive; models tend to output brand priors beginning with "a".

Figure 10: ID: 3
Dominant color: **Yellow**; Shape: **Rectangle/Square**.
GT text: *Artillery*.
Annotator note: high-contrast border.

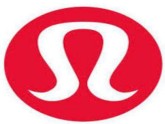

Figure 11: ID: 4
Dominant color: **Black**; Shape: **Complex/Polygon**.
GT text: *HOT WHEELS*.
Annotator note: repeated symmetric arms; models often output well-known brand with similar motif.

Figure 12: ID: 5
Dominant color: **Yellow**; Shape: **Rectangle/Square**.
GT text: *American Standard*.
Annotator note: interior negative space resembles "S"; flagged as *symbol-only*.

Figure 13: ID: 6
Dominant color: **Red**; Shape: **Circle/Ellipse**.
GT text: Ω.
Annotator note: no actual glyphs present.

