# OpenReview forum: "When Symbols Speak: Understanding Logo Triggered Texts in Vision-Language Models"
_ICLR.cc/2026/Conference — ICLR 2026 Conference Withdrawn Submission_

### Official Review · Reviewer_QuUv · 2025-10-26

**Soundness:** 3
**Presentation:** 2
**Contribution:** 1
**Rating:** 4
**Confidence:** 5

**Summary:**

The paper introduces logo hallucination, the phenomenon where Vision-Language Models (VLMs) incorrectly generate brand names or textual outputs from logos that contain no actual text. They apply controlled perturbations and embedding-level diagnostics to demonstrate that hallucination persists under image distortions and originates from specific projector subspaces within the VLM architecture. Targeted ablation of a small number of projector dimensions reduces hallucination by ~30% with minimal OCR accuracy loss.

**Strengths:**

1. Combines taxonomy (text/symbol/hybrid logos), controlled perturbations, and embedding diagnostics in a reproducible and well-defined experimental pipeline.

2. Clarity and presentation: The figures, tables, and structure (bias, perturbation, projector) make the argument cohesive and empirically grounded.

**Weaknesses:**

1. The study is constrained to logo datasets. While logos are a clean diagnostic, it’s unclear whether findings generalize to natural images or scene text. This restriction limits the paper’s general significance.

2. The work is primarily diagnostic rather than methodological. It identifies a failure mode (logo hallucination) and analyzes it well, but does not propose or validate a concrete new training or architectural solution. The projector ablation experiment is an insightful analysis tool, not a deployable method.

3. The link between projector dimensions and hallucination is correlational; ablation reduces the symptom but does not establish a principled mechanism or causal model.

4. The discussion on emotional/value hallucination (luxury, elegance) is anecdotal—no quantification or modeling of this axis.

**Questions:**

Does logo hallucination persist in other symbolic domains (flags, icons, traffic signs)? Could this be a broader symbolic-text entanglement issue?

---

### Official Review · Reviewer_t4Mq · 2025-10-28

**Soundness:** 3
**Presentation:** 2
**Contribution:** 2
**Rating:** 4
**Confidence:** 3

**Summary:**

This paper studies "logo hallucination" in VLMs — where models generate brand names from purely symbolic logos. Through careful experiments across logo types, image perturbations, and projector analysis, the authors show this issue is widespread, robust to input changes, and linked to specific embedding directions. The work offers both a new diagnostic perspective and a practical mitigation path.

**Strengths:**

Logo hallucination is a new and relevant failure mode that hasn't been systematically studied before.

The three-stage framework provides a rigorous, systematic, and multi-faceted investigation into the phenomenon.

**Weaknesses:**

- The paper positions "logo hallucination" as a new and overlooked phenomenon. However, the tendency for models to hallucinate highly correlated labels for salient visual concepts is a well-known issue (e.g., "object hallucination"). The paper would be strengthened by a more explicit discussion of how the demonstrated "logo hallucination" constitutes a meaningfully distinct failure mode, rather than simply being a specific manifestation of the broader object hallucination problem.

- Insufficient Detail on the Hard-60 Subset: More detailed statistics and examples in the main text would help readers understand its composition and the nature of its challenge.

- The current separation of the comprehensive methodology (Section 3) from the experimental results (Section 4) makes the paper occasionally hard to follow. Perhaps restructuring the sections to present each experimental phase alongside its immediate results would improve the narrative flow and readability.

**Questions:**

Please refer to Weaknesses.

---

### Official Review · Reviewer_G9zL · 2025-11-02

**Soundness:** 2
**Presentation:** 3
**Contribution:** 2
**Rating:** 2
**Confidence:** 3

**Summary:**

This paper investigates the phenomenon of logo hallucination in Vision-Language Models (VLMs). The authors discover that VLMs generate brand names even when logos contain no visible text, suggesting that models rely on symbolic priors rather than genuine visual recognition. The study systematically examines this phenomenon, revealing logo hallucination behavior across different models, perturbations, and scenarios.

**Strengths:**

1. Logo hallucination is an overlooked yet significant security risk in VLMs. This problem has practical implications for applications such as brand impersonation detection and fraud prevention.

2. Experiments are conducted across multiple mainstream VLMs to validate the generalizability of the findings.

**Weaknesses:**

1. Insufficient evidence for causal attribution.
- The paper claims that hallucinations stem from "symbolic priors in token embeddings" rather than visual feature extraction problems, but the evidence is insufficient to support this causal inference.
- The paper's logic: Ablating k=32 key dimensions → reduced hallucination rate → concludes these dimensions contain "symbolic priors" → proves it's a token embedding problem. However, this finding can equally be explained as a visual feature representation issue.

2. Lack of mechanistic explanation. While the paper states that logic hallucination stem from "symbolic priors rather than genuine glyph perception," it does not explain how these priors form during training

3. Limited experimental scale. Hard-60 contains only 60 logo samples, potentially insufficient to support strong generalization claims.

**Questions:**

1. Experiments primarily use English brand names. Do logos with text in other languages (e.g., Chinese, Japanese signage) produce similar hallucinations?

2. How was k=32 determined? Was grid search or cross-validation performed? How would different k values (e.g., k=16 or k=64) affect the results?

---

### Note · Authors · 2026-01-26

I have read and agree with the venue's withdrawal policy on behalf of myself and my co-authors.

---

### Meta-Review · Area_Chair_AXm3 · 2025-12-04

**Summary:**

The paper received initial scores of 4, 2, and 4, which is below the expected threshold for acceptance. Notably, the authors did not participate in the rebuttal phase and therefore did not address the reviewers’ concerns regarding the scale of experiments, explanation of the proposed mechanism, implementation details, and ablation studies. After carefully considering the reviewers’ comments and the absence of a substantive rebuttal, I believe the paper is not yet ready for publication and recommend rejection.

**Reviewer Scores:**

None

---

### Decision · Program_Chairs · 2026-01-26

Reject